# Influence of the Casein Genotype on Goat Milk Bioactivity: An In Silico Analysis of the Casein Peptidome

**DOI:** 10.3390/molecules30122601

**Published:** 2025-06-15

**Authors:** Aram Y. Rubio-Reyes, Iván Delgado-Enciso, Eduardo Casas, Estela Garza-Brenner, Ana M. Sifuentes-Rincón

**Affiliations:** 1Laboratorio de Biotecnología Animal, Centro de Biotecnología Genómica, Instituto Politécnico Nacional, Reynosa 88710, Mexico; arubior2300@alumno.ipn.mx; 2Facultad de Medicina, Universidad de Colima, Colima 28040, Mexico; ivan_delgado_enciso@ucol.mx; 3National Animal Disease Center, USDA, ARS, Ames, IA 50010, USA; eduardo.casas@usda.gov; 4Universidad Autónoma de Nuevo León, General Escobedo 66050, Mexico; egarzab@uanl.edu.mx

**Keywords:** caseins, genotypes, bioactive peptides, polymorphism

## Abstract

Goat caseins are highly polymorphic proteins that affect milk functional properties. In this study, an in silico approach was employed to analyze the influence of goat casein allelic variants on the quantity and bioactivity potential of peptides released after enzymatic hydrolysis. The reported protein sequences from the most frequent allelic variants in *Capra hircus* caseins (α-S1, β, α-S2, and κ-casein) were analyzed in the BIOPEP-UWM database to determine the frequency of occurrence of bioactive fragments from each casein. After specific hydrolysis with pepsin, trypsin, and chymotrypsin A, important differences in the peptide profile and bioactivity potential were observed within and between the casein allelic variants. The β-casein A and C alleles, α-S1-casein allele E, and α-S2-casein allele F presented the highest bioactivity potential, and some allele-specific peptides were also released, highlighting the impact of genotype on the predicted bioactivity. The inhibition of angiotensin-converting enzyme (ACE-I) and dipeptidyl peptidase IV (DPP-IV) activities was the most frequent bioactivity of the released peptides, suggesting possible antihypertensive and antidiabetic effects. Once confirmed by experimental studies, the use of goat casein genotyping could direct efforts to enhance the functional quality of goat milk.

## 1. Introduction

Milk proteins are considered reference macronutrients for determining the nutritional quality of dairy products [1]. Caseins have a significant economic impact because of their influence on the quality, quantity, coagulation capacity, and techno-functional properties of proteins, which are key aspects in the production of dairy byproducts [2].

In most mammals, four genes encoding caseins (*CSN1S1* (α-S1), *CSN2* (β), *CSN1S2* (α-S2), and *CSN3* (κ)) have been reported. These genes have highly polymorphic sequences, including single-nucleotide polymorphisms (SNPs), insertions, deletions, and even differences in splicing patterns [2,3]. Casein polymorphisms produce allelic variants that are related to the amount of casein produced; these variants also affect the fat content, micelle size, mineralization, and sensory characteristics of milk [2]. In some ruminants, such as cattle, casein polymorphisms are also related to bioactive peptide production, with the A1 allele of the β-casein gene exhibiting the strongest association. The A1 allele releases the β-casomorphin-7 bioactive peptide [4]. The molecular differentiation between the A1 and A2 alleles has been the basis for the development of new brands in the milk industry [5,6].

Studies of milk-derived bioactive peptides have been conducted via different approaches. Bioinformatic tools such as website-based databases have been widely used as efficient and cost-effective alternatives to search for bioactive peptides from a specific source [7].

In goats, the *CSN1S1* gene has at least twenty known alleles, the *CSN2* gene includes eight alleles, the *CSN1S2* gene consists of fourteen variants, and the *CSN3* gene has at least seventeen known polymorphisms, resulting in twenty-four DNA variants and fifteen protein variants [2,3]. The variability of goat casein genes has been studied worldwide for a variety of goat breeds and locations, and some alleles are usually reported as the most frequent. In the *CSN1S1* gene, the most frequent alleles are A, B, E, and F, with frequencies ranging from 0.5 to 0.70 [2,8,9]. In the *CSN2* gene, the A and C alleles are the most common, with frequencies greater than 0.70 [10]. Alleles A, B, C, and F are the most common (0.40–0.70) in the *CSN1S2* gene [2]. Finally, in the *CSN3* gene, the most frequent alleles are A, B, C, D, and E, with frequencies ranging from 0.20 to 0.70 [2].

Some studies have shown that the enzymatic hydrolysis of goat milk caseins releases bioactive peptides with opioid, immunomodulatory, antimicrobial, antithrombotic, growth-stimulating, and antihypertensive effects [11,12,13]; however, to our knowledge, no studies have associated the content and composition of bioactive peptides derived from the proteolysis of goat milk caseins with their specific allelic variants.

The aim of this study was to predict the genotype-specific bioactivity from goat caseins’ allelic variants through an approach involving the casein allelic sequence data and in silico enzymatic digestion simulations using the BIOPEP-UWM database.

## 2. Results

### 2.1. Analysis of Caseins as a Source of Bioactive Peptides on the Basis of Specific Genotypes

Casein protein sequences were used to determine the parameter A values, and each allele was associated with potential bioactivities on the basis of its amino acid sequence. Differences between alleles of the same type of casein were identified, confirming the effect of allelic variants on the potential bioactivity of caprine casein.

As shown in Table 1, β-casein allele A presented the greatest total frequency of bioactive fragments (ΣA), whereas allele F of α-S1-casein presented the lowest value.

### 2.2. In Silico Proteolytic Release and Prediction of Bioactive Functions

Figure 1 shows a comparison of the specific positions on each allelic protein variant that promotes in silico proteolysis and peptide release via pepsin (EC 3.4.23.1), trypsin (EC 3.4.21.4), and chymotrypsin A (EC 3.4.21.1). All the casein variants analyzed presented different peptide release profiles, with the release peptides varying in quantity, composition, and reported bioactivity. Briefly, α-S1-casein alleles A, B, and E released 48 peptides, whereas allele F released only 27 peptides. The alleles of β, α-S2, and κ-caseins released 53, 51, and 35 peptides, respectively, with variations in peptide composition identified within each allelic variant.

Once the peptides were released, the potential bioactivity index was predicted, and considering a 0.5 threshold value and peptide length of 2–6 amino acids, 35 potential bioactive peptides were found, 32 of which were registered in the BIOPEP-UWM database (Table 2). Regarding their distribution, 14 bioactive peptides were identified in the alleles of α-S1-casein, with distributions of 9, 9, 10, and 6 peptides in alleles A, B, E, and F, respectively. For α-S2-casein, 10 bioactive peptides were found, with 9 present in alleles A, B, and C and 9 present in allele F, differing only by the AGAF peptide in alleles A, B, and C and AGPF in allele F. Finally, in the alleles of β-casein and κ-casein, 9 and 7 bioactive peptides were identified, respectively.

Twenty-eight peptides exhibited two or more bioactive functions. Those released due to amino acid changes resulting from the allelic variant were particularly relevant, as they highlighted the impact of the genotype on the predicted bioactivity of the identified peptides. In the specific case of α-S1-casein, the STF peptide was identified in the F allele and is associated with pancreatic lipase inhibitory activity, a function exclusive to this allele among all analyzed caseins. In the E allele, the amino acid substitution of T to A at positions 209 and 210 in the amino acid sequence releases the AAM peptide, which is considered potentially bioactive (compared with the alleles A and B peptide, TTM). This peptide stands out for its potential multifunctionality, with unique bioactivities such as antibacterial effects, inhibition of D-Ala-D-Ala dipeptidase activity, hypotensive effects, and inhibition of cytosolic alanyl aminopeptidase activity.

In α-S2-casein, an amino acid substitution of A to P at position 134 in allele F (as shown in the sequence alignment) resulted in the AGPF peptide, which exhibited greater bioactivity potential than the AGAF peptide released by alleles A, B, and C. This amino acid change altered the bioactivity profile of the peptides: the AGAF peptide may present effects such as inhibition of alanine carboxypeptidase activity, inhibition of tripeptidyl peptidase II activity, and CaMPDE inhibition, the last of which is exclusive to this allele. On the other hand, the AGPF peptide has a distinct bioactivity profile, including antiamnestic activity, ACE2 and neprilysin inhibition, antithrombotic activity, and regulatory functions.

Finally, Figure 2 illustrates the expected bioactivity potential of all the goat casein alleles on the basis of peptides released through in silico hydrolysis with pepsin, trypsin, and chymotrypsin. The color variations in the heatmap clearly show that the predominant effects of all the caseins include the inhibition of angiotensin-converting enzyme (ACE-I) and dipeptidyl peptidase IV (DPP-IV-I) activities. Differences in the bioactivity of each allele within a specific casein were also observed, and the alleles with lower expected bioactivity potential are represented by lighter tones. For example, in α-S1-casein, a reduction in ACE, DPP-IV, dipeptidyl peptidase III, and renin inhibition was observed in some alleles. Similarly, in α-S2-casein, lower antioxidant activity and reduced dipeptidyl peptidase III inhibition were identified. In contrast, in β-casein and κ-casein, no significant variations in bioactivities were found among the alleles.

After evaluating the bioactivity potential of each casein allele, it could be inferred that β-casein alleles A and C have the potential to exhibit the highest bioactivity, with a greater probability of displaying a more comprehensive bioactive profile. The α-S1-casein allele E has the potential to exhibit greater bioactivity than alleles A, B, and F do, with the latter being the least functional of all caseins. In α-S2-casein, allele F stands out as the most functional allele compared with alleles A, B, and C. For κ-casein, all the alleles share the same theoretical bioactivity profile.

## 3. Discussion

The results obtained in this study allowed the identification of changes in bioactive peptide release due to variations in the protein sequences of allelic variants of caseins. In recent decades, several studies have reported the release of bioactive peptides from goat milk and goat dairy products [13,14]. However, the effects of allelic variants on the release of these peptides in goat caseins have not been evaluated in vivo or in silico. Here, after the total frequency of bioactive fragment occurrence (ΣA) was calculated, differences were identified among the four types of caseins, as well as among alleles of the same casein type, indicating a high probability that genotype significantly influences the release of bioactive peptides.

Different studies in cattle and other species have shown that a higher ΣA value indicates a greater probability of bioactive peptides being released from the analyzed sequence [15]. Lin et al. [16] and Gu et al. [17] evaluated the potential for bioactive peptide release in yak and buffalo milk, respectively, via the A parameter. In both cases, β-casein presented higher ΣA values than did α-S1, α-S2, and κ-casein, similar to the results obtained in this study. However, the ΣA value in the goat caseins analyzed in this study was even greater than that reported for yak and buffalo milk caseins, suggesting that goat caseins have greater potential for releasing bioactive peptides.

The release of bioactive peptides from proteins through enzymatic hydrolysis depends on the specificity of proteases in recognizing and cleaving peptide bonds within the protein. The type of enzymes used and the hydrolysis conditions vary depending on the nature of the precursor protein, thereby influencing the release and activity of bioactive peptides [18,19,20]. In this context, the enzymatic hydrolysis of dairy proteins has been widely applied to improve their solubility in water and release peptides with potential biological effects [21]. Different studies in goats and other species have proposed the use of pepsin, trypsin, and chymotrypsin as hydrolytic proteases because of their specificity for cleaving peptide bonds. Trypsin is specific for lysine and arginine residues, whereas chymotrypsin is specific for aromatic amino acid chains such as tryptophan, phenylalanine, and tyrosine [7,22]. Pepsin is the principal enzyme involved in protein digestion in the gastric phase and is a widely used digestive protease for milk and other food proteins [23].

The in silico analysis of pepsin, trypsin, and chymotrypsin confirmed that goat milk caseins, depending on their genotype, are suitable precursors for the release of bioactive peptides. The predictive tool allowed the identification of differences in the quantity and composition of the peptides released, highlighting the impact of genetic variations on the peptide profile.

A clear example of this variability was observed in α-S1-casein hydrolysis. In this study, the E allele presented the most extensive peptide profile, whereas the F allele presented a reduced profile because of the lack of recognition sites. This could be explained by the loss and modification of amino acids reported for this allele due to a cytosine deletion in exon 9, along with 11- and 3-base pair insertions in intron 9, of the *CSN1S1* gene. These alterations cause frameshifts that eliminate exons 9, 10, and 11 [2,24,25]. Consequently, this allele releases the peptide STF, which exhibits pancreatic lipase inhibitory activity, suggesting a possible antiobesity effect [26].

Moreover, single or double amino acid substitutions in casein genes modified the bioactivity of the released peptides. In α-S2-casein, the substitution of A for P at position 134 of the F allele resulted in the peptide AGPF, whereas in the A, B, and C alleles, the peptide AGAF was released. This amino acid change produced different functional profiles; the AGAF peptide has unique CaMPDE inhibitory activity, which could be useful in the treatment of atherosclerosis and cancer [27], whereas in α-S1-casein, the replacement of the amino acid sequence TTM (A allele) with AAM at positions 209 and 210 in the E allele converted a previously nonbioactive fragment into a peptide with a high probability of bioactivity. This peptide exhibited unique bioactivities among the four types of casein, such as antibacterial activity, hypotensive effects, and inhibition of cytosolic alanyl aminopeptidase and D-Ala-D-Ala dipeptidase activities, the last of which potentially contribute to antimicrobial effects [28].

The high probability of the peptides inhibiting angiotensin-converting enzyme (ACE-I) and dipeptidyl peptidase IV (DPP-IV-I) activities could provide opportunities for the development of functional foods with antihypertensive and antidiabetic effects, respectively [29,30,31].

According to the BIOPEP-UWM database, the enzymes used in this study have a preferential cleavage site within the casein sequences. Pepsin cleaves at the C-terminal end after residues such as phenylalanine (Phe) and leucine (Leu), trypsin cleaves at the C-terminal end after residues such as lysine (Lys) and arginine (Arg), while chymotrypsin cleaves at the C-terminal end after residues such as tyrosine (Tyr), tryptophan (Trp), phenylalanine (Phe), leucine (Leu), asparagine (Asn), histidine (His), and methionine (Met); at the N-terminal end, the enzyme preferentially cleaves isoleucine (Ile).

Other in silico studies have evaluated combinations of different enzymes to release bioactive peptides from goat caseins. Wu et al. [32] used the BIOPEP-UWM database to simulate the hydrolysis of goat milk β-casein with pepsin (EC: 3.4.23.1), chymotrypsin A (EC: 3.4.21.1), and trypsin (EC: 3.4.21.4), identifying 51 peptides. In our study of β-casein, we identified 53 peptides, confirming that the high polymorphism of caseins may modify enzyme recognition sites.

In this study, we identified a greater number of bioactivities for the allelic variants of β-casein than for those of α-S1-casein, α-S2-casein, and κ-casein. According to reported studies, most of the biofunctionalities identified in milk from different species have been associated primarily with peptides derived from β-casein, followed by these three caseins, which is consistent with the results obtained in this in silico study [13]. This could be related to the structure of β-casein and its susceptibility to enzymatic hydrolysis, facilitating the release of peptides with well-documented biological effects. Our results suggest that while β-casein, α-S1, and α-S2 may represent promising sources of bioactive peptides, experimental studies are needed to confirm their functionality and compare their impact with peptides derived from β-casein.

Integrating genomic and bioinformatic tools represents a promising strategy for the development of dairy products with health benefits. However, this type of analysis has limitations. In silico proteolysis does not guarantee that the predicted peptides can be experimentally reproduced, as proteins may not undergo complete hydrolysis [33], and also, the predictive power of the method depends on the cleavage sites reported for proteases, which may not accurately reflect their actual behavior, as the bioactivities of peptides are predicted based on sequence similarity to previously characterized fragments [16]. Despite these considerations, the obtained results have important practical applications in the discovery of bioactive peptides, such as directing more cost-effective experimental designs to test and validate the in silico results.

Future work could include the in vitro hydrolysis of genotype-specific precursor proteins or even the direct synthesis of the most promising bioactive peptides, followed by experimental evaluation of their biological effects [34].

Finally, considering that casein allelic variant genotyping is a practical tool available for most producers, once the bioactive potential of the casein allelic variants is confirmed, genotyping goat populations could guide breeding strategies aimed at enhancing the functional properties of dairy products by selecting those carriers of the casein haplotypes that have higher bioactive potential.

## 4. Materials and Methods

### 4.1. Analysis of Capra Hircus Caseins as a Source of Bioactive Peptides

The protein sequences of *Capra hircus* caseins were selected from the NCBI database (https://www.ncbi.nlm.nih.gov/ accessed on 7 April 2025). Because some protein sequences from allelic variants reported as high-frequency alleles in goats are not available in the NCBI database, hypothetical sequences were generated on the basis of the reference alleles of each casein and by substituting the amino acid composition at the positions described in the literature [2]. The studied allelic variants were as follows: for α-S1 casein, alleles A (NP_001424592.1, reference allele), B (hypothetical sequence), E (X72221.1), and F (AJ504711.2); for β-casein, alleles A (AJ011018.3) and C (AY563136.1, reference allele); for α-S2 casein, alleles A (XP_013820127.2, reference allele), B (hypothetical sequence), and C (hypothetical sequence) and F (AJ289716.1); and for κ-casein, alleles A (AF485339.1), B (AF485340.1, reference allele), C (AF485341.1), D (AY090465.1), and E (AF486523.1). Alignment, comparison, and analysis of casein protein sequences were performed via MEGA XI software v 11.0.13 [35]. Once aligned, positions with amino acid changes due to allelic variations in each type of casein were identified. The workflow of the analysis is shown in Figure 3.

The potential of the selected casein allele sequences to release bioactive peptides was determined via the BIOPEP-UWM database (https://biochemia.uwm.edu.pl/biopep-uwm/ accessed on 22 April 2025), which contains 5213 known bioactive peptides and 93 biological activities. The BIOPEP-UWM not only simulates proteolytic digestion but also provides curated information from the scientific literature about peptide sequences, protein sources, extraction methods, and evidence of bioactivity. It also integrates data from other specialized databases [36,37].

Using the “calculations” tool available in this database, the A parameter was determined (a quantitative value that characterizes proteins as potential precursors of bioactive peptides). This parameter represents the frequency of the occurrence of bioactive fragments within a protein sequence, considering all bioactive peptides present and their possible bioactivity. This parameter was calculated via the following equation:A = *a*/*N*
where
*a* = Number of fragments with a specific activity in a protein sequence;*N* = Total number of amino acid residues in the protein.


The total frequency of bioactive fragments in the sequences of each casein allele (α-S1, β-casein, α-S2, and κ-casein) (∑A) was subsequently determined. This is an unweighted summed value that indicates the number of bioactive fragments (with any biological activity reported in the BIOPEP-UWM database) identified in the analyzed sequence [16,36].

### 4.2. In Silico Proteolytic Release and Prediction of Bioactive Functions

For proteolytic release, in silico hydrolysis of each casein/allelic sequence was executed via the BIOPEP-UWM database with the “ENZYME(S) ACTION” tool. The release of bioactive peptides was simulated via the simultaneous action of pepsin (EC 3.4.23.1), trypsin (EC 3.4.21.4) and chymotrypsin A (EC 3.4.21.1) [36,37]. Cleavage sites were assigned based on the default specificity parameters provided by the BIOPEP-UWM platform for each enzyme. These rules define the expected hydrolysis patterns according to enzymatic cleavage preferences previously determined. The simulation considered overlapping peptides resulting from successive cleavage events, which were included in the output.

The potential bioactivity index of the fragments released by proteolysis was evaluated via PeptideRanker software http://distilldeep.ucd.ie/PeptideRanker/ (accessed on 24 April 2025), which predicts how likely the peptide is to be bioactive, on the basis of a novel N-to-1 type neural network. PeptideRanker assigns score values ranging from 0 to 1, where a value greater than 0.5 indicates a high probability of bioactivity; only peptides that exceeded this threshold were considered for evaluating their potential biological activity profile via the “profiles of potential biological activity” tool available in the BIOPEP-UWM bioactive peptide database. The profile of potential biological activity refers to the nature and location of bioactive fragments within a protein or peptide sequence. This concept is based on the premise that the same bioactive fragment, particularly if it is short (2–3 amino acid residues), is not unique to a single protein but can be found in multiple sequences, forming what are known as conserved or common subsequences [36,38,39].

A heatmap was generated in RStudio V.2024.9.1.394 [40] to visualize the assigned bioactivities from each casein allele on the basis of the number of identified bioactive peptides. As different peptides were found to have the same bioactivity, for each gene and allelic variant, a matrix containing the sum values of each bioactivity was then created, and the color gradient in the heatmap was proportional to the intensity of the activity, where white indicates the absence of activity, whereas colored shading represents different levels of bioactive activity.

## 5. Conclusions

In silico analysis revealed that the most frequent allelic variants of the four casein genes impact the bioactive properties of goat milk, altering the number and composition of peptides released after hydrolysis with different digestive enzymes.

Specific and unique bioactive peptides were found to be α-S2-casein F allelic variants and between α-S1-casein E and F allelic variants. Compared with the other analyzed alleles of the casein genes, the β-casein A and C alleles, α-S1-casein E allele, and α-S2-casein F allele presented greater bioactivity potential.

Additional experimental studies are needed to validate the in silico results and, thus, use casein genotyping to select carriers with allelic variants of goat milk caseins with favorable bioactivities.

## Figures and Tables

**Figure 1 molecules-30-02601-f001:**
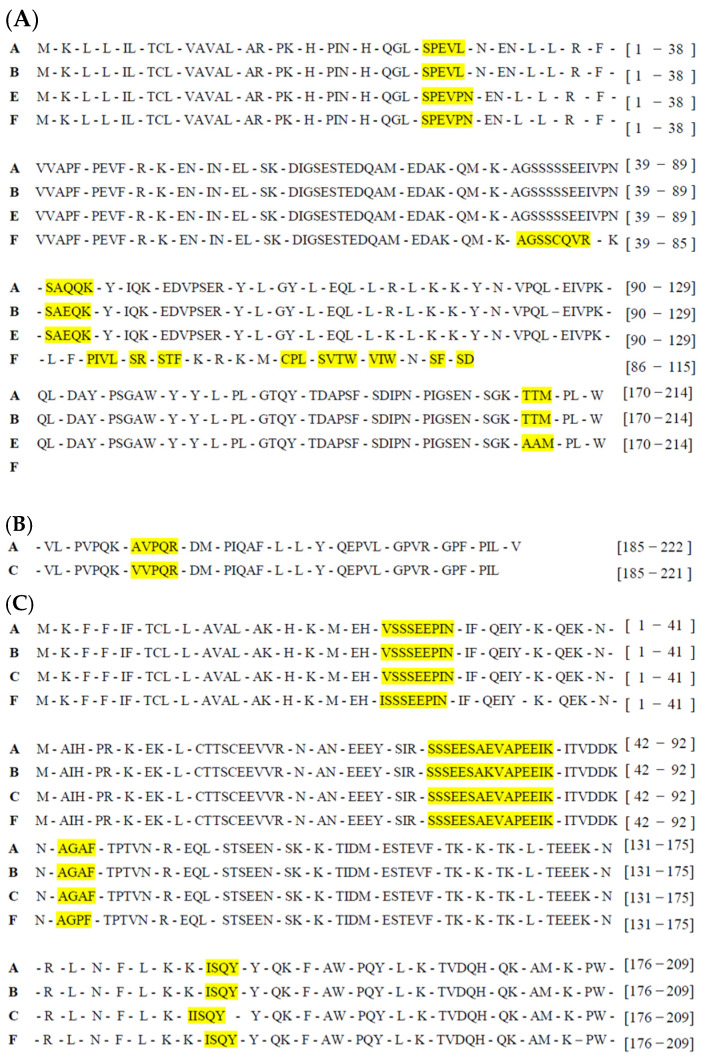
Comparison of in silico proteolysis results using pepsin, trypsin, and chymotrypsin in goat casein alleles. Only the positions where there were differences in the quantity and type of released peptides occur are shown: (**A**) α-S1-casein alleles, (**B**) β-casein alleles, (**C**) α-S2-casein alleles, and (**D**) κ-casein alleles. Allele-specific peptides are highlighted in yellow. A comparison of the complete proteolysis profile for each casein is presented in Appendix B.

**Figure 2 molecules-30-02601-f002:**
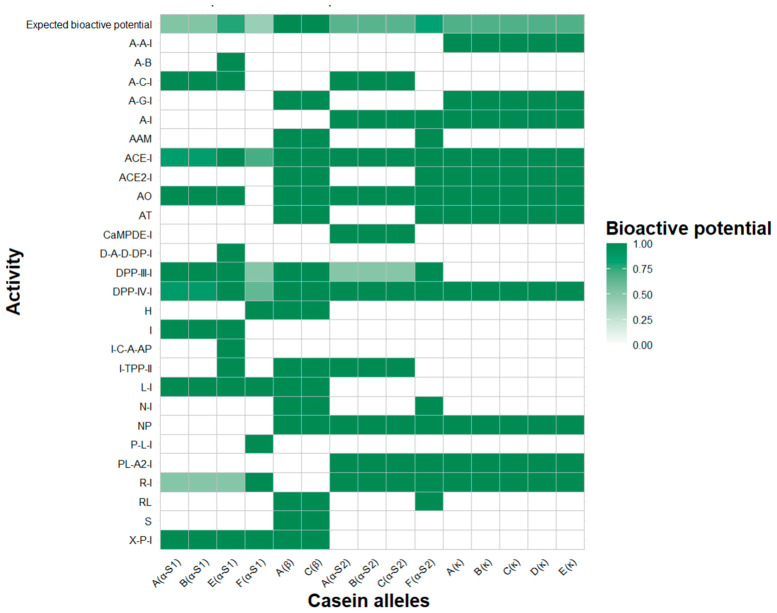
Expected bioactive potential of goat casein alleles, where white indicates the absence of activity, while colored shades represent different levels of bioactive activity. A(α-S1): α-S1-casein allele A. B(α-S1): α-S1-casein allele B. E(α-S1): α-S1-casein allele E. F(α-S1): α-S1-casein allele F. A(β): β-casein allele A. C(β): β-casein allele C. A(α-S2): α-S2-casein allele A. B(α-S2): α-S2-casein allele B. C(α-S2): α-S2-casein allele C. F(α-S2): α-S2-casein allele F. A(κ): κ-casein allele A. B(κ): κ-casein allele B. C(κ): κ-casein allele C. D(κ): κ-casein allele D. E(κ): κ-casein allele E. ACE inhibitor: ACE-I. ACE2 inhibitor: ACE2-I. Alanine carboxypeptidase inhibitor: A-C-I. Alpha-amylase inhibitor: A-A-I. Alpha-glucosidase inhibitor: A-G-I. Anti inflammatory: A-I. Antiamnestic: AAM. Antibacterial: A-B. Antioxidative: AO. Antithrombotic: AT. CaMPDE inhibitor: CaMPDE-I. D-Ala-D-Ala dipeptidase inhibitor: D-A-D-DP-I. Dipeptidyl peptidase III inhibitor: DPP-III-I. Dipeptidyl peptidase IV inhibitor: DPP-IV-I. Hypotensive: H. Inhibitor: I. Inhibitor of cytosol alanyl aminopeptidase: I-C-A-AP. Inhibitor of tripeptidyl peptidase II: I-TPP-II. Lactocepin inhibitor: L-I. Neuropeptide: NP. Neprilysin inhibitor: N-I. Pancreatic lipase inhibitor: P-L-I. Phospholipase A2 inhibitor: PL-A2-I. Regulating: RL. Renin inhibitor: R-I. Stimulating: S. Xaa-pro inhibitor: X-P-I.

**Figure 3 molecules-30-02601-f003:**
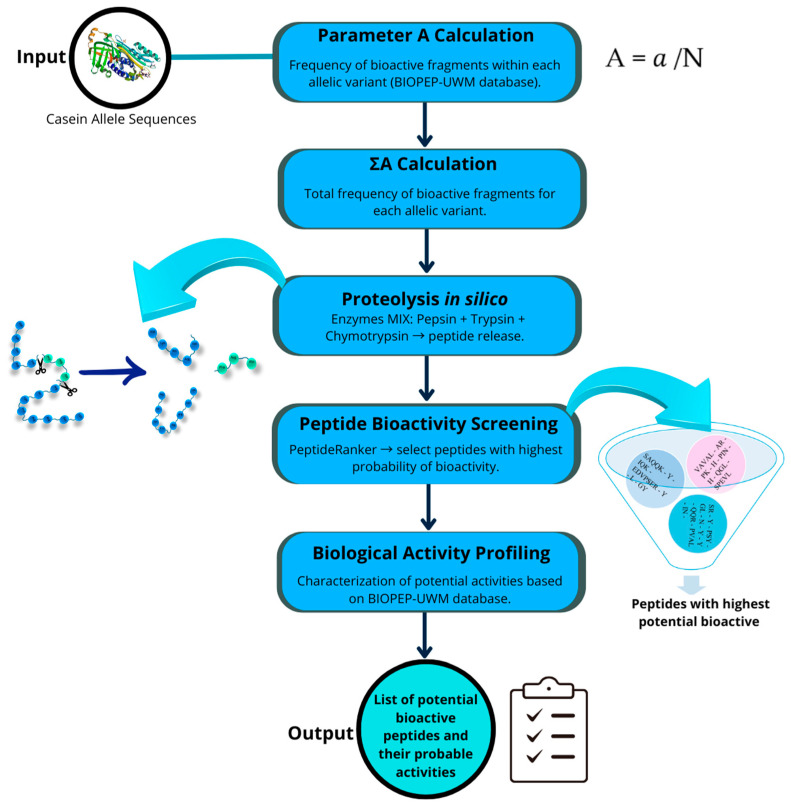
Workflow of the in silico analysis of bioactive peptide release from casein allelic variants.

**Table 1 molecules-30-02601-t001:** Total frequency of the detected functional activities of bioactive fragments (∑A) in the most frequent allelic variants of goat caseins.

Casein Variant	ΣA *
α-S1-casein	
A	2.1447
B	2.1541
E	2.1216
F	1.6351
β-casein	
A	2.4500
C	2.4429
α-S2-casein	
A	1.9959
B	1.9735
C	1.9645
F	2.0094
κ-casein	
A	2.1914
B	2.2038
C	2.2161
D	2.2286
E	2.2223

* A full description of each counted activity used to determine ΣA is presented in Appendix A.

**Table 2 molecules-30-02601-t002:** Bioactive fragments released through in silico proteolysis of goat casein alleles using pepsin, trypsin, and chymotrypsin.

Peptide	Peptide Ranker *	Bioactive Activity	α-S1-Casein				β-Casein		α-S2-Casein				κ-Casein				
			A	B	E	F	A	C	A	B	C	F	A	B	C	D	E
QGL	0.533495	ACE-I; DPP-IV-I	✓	✓	✓	✓											
QM	0.607122	-	✓	✓	✓	✓											
STF	0.58977	ACE-I; H; DPP-IV-I; DPP-III-I; R-I; P-L-I				+											
CPL	0.916551	ACE-I; DPP-IV-I; X-P-I; L-I				+											
VIW	0.610494	ACE-I; DPP-IV-I				+											
SF	0.948796	ACE-I; DPP-IV-I; R-I				✓							✓	✓	✓	✓	✓
GY	0.741592	ACE-I; DPP-IV-I; DPP-III-I	✓	✓	✓												
SM	0.628268	DPP-III-I	✓	✓	✓												
QPM	0.810167	ACE-I; DPP-IV-I	✓	✓	✓												
PQL	0.527882	ACE-I; DPP-IV-I	✓	✓	✓												
QF	0.946135	DPP-IV-I; R-I	✓	✓	✓				✓	✓	✓	✓	✓	✓	✓	✓	✓
PSGAW	0.870583	ACE-I; AO; I; DPP-IV-I; A-C-I	✓	✓	✓												
PL	0.811148	ACE-I; DPP-IV-I; X-P-I; L-I	✓	✓	✓		✓	✓									
AAM	0.55858	ACE-I; A-B; H; DPP-IV-I; I-TPP-II; I-C-A-AP; D-A-D-DP-I			+												
ACL	0.746733	-					✓	✓									
PF	0.99343	DPP-IV-I; DPP-III-I; ACE2-I					✓	✓									
PVEPF	0.604782	ACE-I; AO; DPP-IV-I; A-G-I; DPP-III-I; ACE2-I; N-I					✓	✓									
QPPQPL	0.823819	ACE-I; DPP-IV-I; A-G-I; X-P-I; L-I					✓	✓									
DM	0.607466	ACE-I					✓	✓									
PIQAF	0.749093	ACE-I; DPP-IV-I; I-TPP-II					✓	✓									
GPF	0.989324	AAM; ACE-I; AT; RL; AO; H; DPP-IV-I; DPP-III-I; ACE2-I; N-I					✓	✓									
PIL	0.641797	ACE-I; S; NP; DPP-IV-I					✓	✓									
IF	0.949173	ACE-I							✓	✓	✓	✓					
PR	0.787626	ACE-I; DPP-III-I							✓	✓	✓	✓					
PY	0.736696	NP; A-I; DPP-IV-I; PL-A2-I							✓	✓	✓	✓	✓	✓	✓	✓	✓
PW	0.992911	AO; DPP-IV-I							✓	✓	✓	✓					
AGAF	0.873661	ACE-I; AO; DPP-IV-I; CaMPDE-I; I-TPP-II; A-C-I							✓	✓	✓						
AGPF	0.969222	AAM; ACE-I; AT; RL; AO; DPP-IV-I; DPP-III-I; ACE2-I; N-I										+					
AW	0.9669	ACE-I; AO; DPP-IV-I							✓	✓	✓	✓					
AM	0.74549	-							✓	✓	✓	✓					
AIPY	0.613395	ACE-I; NP; A-I; DPP-IV-I; PL-A2-I							✓	✓	✓	✓					
GL	0.808777	ACE-I; DPP-IV-I											✓	✓	✓	✓	✓
QW	0.928524	DPP-IV-I											✓	✓	✓	✓	✓
PH	0.541688	ACE-I; DPP-IV-I; AO											✓	✓	✓	✓	✓
AIPPK	0.575607	A-A-I; ACE-I; AT; AO; A-I; DPP-IV-I; A-G-I; ACE2-I											✓	✓	✓	✓	✓

* Full list of the bioactive potential index is presented in Appendix C. ✓: Peptide released. +: Potentially bioactive peptide released on a single allele. ACE inhibitor: ACE-I. ACE2 inhibitor: ACE2-I. Alanine carboxypeptidase inhibitor: A-C-I. Alpha-amylase inhibitor: A-A-I. Alpha-glucosidase inhibitor: A-G-I. Anti inflammatory: A-I. Antiamnestic: AAM. Antibacterial: A-B. Antioxidative: AO. Antithrombotic: AT. CaMPDE inhibitor: CaMPDE-I. D-Ala-D-Ala dipeptidase inhibitor: D-A-D-DP-I. Dipeptidyl peptidase III inhibitor: DPP-III-I. Dipeptidyl peptidase IV inhibitor: DPP-IV-I. Hypotensive: H. Inhibitor: I. Inhibitor of cytosol alanyl aminopeptidase: I-C-A-AP. Inhibitor of tripeptidyl peptidase II: I-TPP-II. Lactocepin inhibitor: L-I. Neuropeptide: NP. Neprilysin inhibitor: N-I. Pancreatic lipase inhibitor: P-L-I. Phospholipase A2 inhibitor: PL-A2-I. Regulating: RL. Renin inhibitor: R-I. Stimulating: S. Xaa-pro inhibitor: X-P-I.

## Data Availability

Data available on request from the authors.

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
