# Peer review of "Influence of the Casein Genotype on Goat Milk Bioactivity: An In Silico Analysis of the Casein Peptidome"

_molecules, 2025, doi:10.3390/molecules30122601_

Round 1
Reviewer 1 Report
Comments and Suggestions for Authors
This research is entirely dependent on bioinformatic data analysis and is dedicated to the study of the potential content of biologically active peptides formed during hydrodysis of different allelic variants of goat milk caseins. The research topic is of particular pertinence, especially in light of the increasing global consumption of "non-cow" milk.
However, given that this is a purely theoretical work, it would be beneficial to analyse a more extensive array of data, encompassing a greater number of peptides, alleles, and proteases (the current paper presents 10 alleles, 35 peptides, and 2 proteases).
- In the present study, peptides ranging in length from 2 to 6 amino acid residues were included (line 90).
Concurrently, the papers to which you refer (references #10 and 12) contain a substantial number of biologically active peptides with confirmed "useful" activity of much larger sizes (up to 25 amino acid residues). It is therefore questioned, as to why the study was limited to short peptides only.
- The Materials and Methods section, as well as the text of the article itself, do not make it clear how the theoretical proteolysis of caseins was performed by trypsin and chymotrypsin A. It is not evident whether the presented final set of obtained peptides is the result of calculation of the joint (simultaneous) action of these two enzymes or whether the calculation was performed for each protease separately and then combined. In in vivo conditions, both enzymes function concomitantly. In order to model real conditions, it would be beneficial to incorporate data for other digestive proteases, such as pepsin.
- In Conclusion, it is recommended, that the authors provide their own suggestions for the practical utilization of the data obtained by them. For instance, the selection of goat breeds based on the allelic composition of their caseins.
In addition, the quality of the drawings has been the subject of critique. The resolution of the images is substandard. The amino acid sequences of peptides in Figure 1 are rendered illegible.
Author Response
Reviewer: This research is entirely dependent on bioinformatic data analysis and is dedicated to the study of the potential content of biologically active peptides formed during hydrolysis of different allelic variants of goat milk caseins. The research topic is of particular pertinence, especially in light of the increasing global consumption of "non-cow" milk. However, given that this is a purely theoretical work, it would be beneficial to analyze a more extensive array of data, encompassing a greater number of peptides, alleles, and proteases (the current paper presents 10 alleles, 35 peptides, and 2 proteases).
Authors: In our analysis the included alleles were selected on the basis of the reported protein sequences in order to give more certainty and precision to the peptidome analysis, however, in order to present a more complete vision about the potential of these allelic variants in peptide bioactive liberation, and attending the reviewers concerns, we decide to complete our analysis including the more frequent missed alleles, to do it, we analyzed them as hypothetical protein sequences, as it is described in L269-L272.
Here it is important to remark that even the CSN1S1 gene reports twenty alleles, the CSN2 gene eight alleles, CSN1S2 gene fourteen variants, and the CSN3 gene has twenty-four variants, we focused on those reported as a worldwide more frequent because, we consider that from an practical point of view, in most of goats herds they will have a real and significant impact on milk composition and, consequently, on the bioactive potential of the derived peptides. Most of the allelic variants with low frequency are considered as rare variants, their presence in commercial goat populations is very limited, so their exclusion in the analysis could have minimal impact from a practical point of view.
About the digestive enzymes, we primary selected trypsin and chymotrypsin because they have been described as highly specific proteases due to their specificity for cleaving peptide bonds. Trypsin is specific for lysine and arginine residues, while chymotrypsin is specific for aromatic amino acid chains like tryptophan, phenylalanine, and tyrosine. However, to consider both reviewers’ suggestions we have decided also to include the pepsin in our analysis. Pepsin has been described as the principal enzyme involved in protein digestion in the gastric phase. It is a reference as a digestive protease for milk and other food proteins, their use in bioactive peptides liberation from goat milk has been widely documented (please see L194-L200 for a more detailed description about the protease’s selection), so, its inclusion clearly completes our study.
Now our study array encompasses the most frequent allelic variants in goat i.e. 15 alleles, three proteases, and the analysis of 35 peptides with the highest bioactivity probability. In this last regard, is interesting to remark that the inclusion of pepsin in the enzymes did not change the number or composition of the analyzed peptides.
Reviewer: In the present study, peptides ranging in length from 2 to 6 amino acid residues were included (line 90). Concurrently, the papers to which you refer (references #10 and 12) contain a substantial number of biologically active peptides with confirmed "useful" activity of much larger sizes (up to 25 amino acid residues). It is therefore questioned, as to why the study was limited to short peptides only.
Authors: As is stated in L316-L320, “the PeptideRanker assigns score values ranging from 0 to 1, where a value greater than 0.5 indicates a high probability of bioactivity”, in our study even peptides larger to 2 to 5 aminoacidic residues were found (Please see Appendix C, mentioned in L144 of Table 2), we used this threshold to select the studied peptides, thus as it is described in L319-L322, “only peptides that exceeded this threshold were considered for evaluating their potential biological activity profile using the "profiles of potential biological activity" tool available in the BIOPEP-UWM bioactive peptide database, all these peptides ranging in shorth length amino acid residues. We also include a more precise definition of the profile of potential biological activity, please see L322-L327.
Reviewer: In the Materials and Methods section, as well as the text of the article itself, do not make it clear how the theoretical proteolysis of caseins was performed by trypsin and chymotrypsin A. It is not evident whether the presented final set of obtained peptides is the result of calculation of the joint (simultaneous) action of these two enzymes or whether the calculation was performed for each protease separately and then combined.
Authors: We appreciate the observation, a description about how theoretical proteolysis was achieved is described in section 4.2. In silico proteolytic release and prediction of bioactive functions of material and methods section, please see L311-314. In these lines we also specify that resulting peptides are from the mix of the three enzymes. Now we also include a figure to present a clearer workflow (Please see figure 3)
Reviewer: In vivo conditions, both enzymes function concomitantly. In order to model real conditions, it would be beneficial to incorporate data for other digestive proteases, such as pepsin.
Authors: We agree with revisor´s suggestion, as we already commented, in the analysis now we include the results of the pepsin digestion.
Reviewer: In Conclusion, it is recommended, that the authors provide their own suggestions for the practical
utilization of the data obtained by them. For instance, the selection of goat breeds based on the allelic composition of their caseins.
Authors: In L339-L345, we conclude that once validated the bioactive peptide production from each allele, we recommend the integration of genotyping as a practical tool for genetic selection in breeding programs. In particular, the selection of animals carrying the β-casein A and C alleles, the α-S1-casein E allele, and the αS2-casein F allele is suggested, given their potential to yield peptides with high bioactivity. This conclusion is based on discussion please see L250-L256.
Reviewer: In addition, the quality of the drawings has been the subject of critique. The resolution of the images is substandard. The amino acid sequences of peptides in Figure 1 are rendered illegible.
Authors: We verified the figures’ quality and actually they fulfill the requirements of the journal, we think that there was a problem in the pdf file, we hope the current file had the requested quality.
Reviewer 2 Report
Comments and Suggestions for Authors
This work is a typical bio-info study. However, as far as I can see, this work is far from complete because its data base is too small to give a reasonable suggestion. Also, different methods should be used to verify the results. Besides, I think this manuscript is not very suitable for Molecules, as I understand, there are subtle but important difference between bio-info and computational chemistry.
- The figures are so blurred that I cannot read them easily. The figs 1 and 2 are important results, but I cannot read them. What is more, even the MDPI logo is blurred.
- Ln. 275, are there any weights for different components when computing Sigma A. Why?
- I understand that there are some data unavailable in teh database. But does it mean that the conclusion of the current work is not solid, because what is missed is alleles of high frequency in goats?
- The authors should consider to justify their methodology. They do have some description of the methods used, but they should also explain why such methods are chosen. E.g., those methods based on neural networks highly relies on the database used when training the network. The authors should tell the readers the reason why should we trust the algorithm.
Author Response
Reviewer: This work is a typical bioinfo study. However, as far as I can see, this work is far from complete because its data base is too small to give a reasonable suggestion. Also, different methods should be used to verify the results. Besides, I think this manuscript is not very suitable for Molecules, as I understand, there are subtle but important difference between bioinfo and computational chemistry.
Authors: Yes indeed, our study is aimed to use bioinformatic tools based on database analysis to answer a specific but important question about the goat-milk derived bioactive peptides, i.e the influence of goat´s casein allelic variants. We now include more information about how different approaches have been used to explore the production of bioactive peptides liberation from different sources (please see L45-L47). All these methods have been successfully used to direct experimental designs and to do more time and cost-effective analysis of bioactive peptides production (Wu et al., 2023*). Because of this we believe that this manuscript is appropriate for publication in the Special Issue “Bioactive Compounds from Functional Foods, 2nd Edition," of Molecules, the results we found are novel and significant, not only by the prediction of its biological implications in milk goat biofunctionality, but also to support further research to test and explore its application to practical breeding schemes (please see the discussion in L250-L256).
* Wu, Y.; Zhang, J.; Mu, T.; Zhang, H.; Cao, J.; Li, H.; Zhao, K. Selection of goat β-casein derived ACE-inhibitory peptide SQPK and insights into its effect and regulatory mechanism on the function of endothelial cells. Int. J. Biol. Macromol. 2023, 253, 127312. https://doi.org/10.1016/j.ijbiomac.2023.127312
Reviewer: The figures are so blurred that I cannot read them easily. The figs 1 and 2 are important results, but I cannot read them. What is more, even the MDPI logo is blurred.
Authors: We verified the uploaded figures’ quality and actually they fulfills the requirements of the journal, we think that there was a problem in the pdf file, we hope the current file had the requested quality.
Reviewer: Ln. 275, are there any weights for different components when computing Sigma A. Why?
Authors: No, as now is specified (please see L305-L309) ∑A, is an unweighted value considering all identified bioactive fragments.
Reviewer: I understand that there are some data unavailable in the database. But does it mean that the conclusion of the current work is not solid, because what is missed is alleles of high frequency in goats?
Authors: To present a more complete vision about the potential of these allelic variants in peptide bioactive liberation, and attending the reviewers’ concerns, we decide to complete our analysis including the more frequent missed alleles; to do it, we analyzed them as hypothetical protein sequences, as it is described in L269-L272.
Here it is important to remark that even the CSN1S1 gene reports twenty alleles, the CSN2 gene eight alleles, CSN1S2 gene fourteen variants, and the CSN3 gene has twenty-four variants, we focused on those reported as a worldwide more frequent because, we consider that from an practical point of view, in most of goats herds they will have a real and significant impact on milk composition and, consequently, on the bioactive potential of the derived peptides. Most of the allelic variants with low frequency are considered as rare variants, their presence in commercial goat populations is assumed as very limited, so their exclusion in our analysis could have no impact from a practical point of view.
Reviewer:The authors should consider to justify their methodology. They do have some description of the methods used, but they should also explain why such methods are chosen. E.g., those methods based on neural networks highly relies on the database used when training the network*. The authors should tell the readers the reason why should we trust the algorithm.
Authors: Undoubtedly, advances in novel computational models based in machine learning, artificial intelligence, neural and convolutional networks, are revolutionizing and complementing the bioactive peptides research mainly in drugs design. We consider that all these new bioinformatic tools are still in development and each research focused on using it, could be considered as a model itself to do bioactive peptides predictions. Certainly, exploring the potential of these tools in milk-derived peptides could be interesting as a further task.
Here we have decided to use a conventional bioinformatic tool to address the specific question of our study. As has been reviewed in some reports, at least six to seven different website based Databases** could be used for processing proteolytic enzyme analysis (Kashung and Karuthapandian, 2025**). The BIOPEP-UWM database(http://www.uwm.edu.pl/biochemia/index.php/pl/biopep), was selected because it could be considered as a reference database in the research on biologically active peptides, and since it was reported, it has been widely used in food and nutrition science. As we now stated in L290-L293, BIOPEP-UWM not only simulates proteolytic digestion but also provides curated information from scientific literature about peptide sequences, protein sources, extraction methods, and evidence of bioactivity, but also integrates data from other specialized databases.
The reliability of BIOPEP-UWM is supported by its validation in numerous scientific studies, its continuous updates, and its recognition in international academic platforms (FitzGerald et al., 2020***; Zhenghui et al., 2025****).
** Kashung, P.; Karuthapandian, D. Milk-derived bioactive peptides. Food Prod. Process and Nutr. 2025, 7, 6. https://doi.org/10.1186/s43014-024-00280-2
***FitzGerald, R. J.; Cermeño, M.; Khalesi, M.; Kleekayai, T.; Amigo-Benavent, M. Application of in silico approaches for the generation of milk protein-derived bioactive peptides. J. F. F. 2020, 64, 103636 . https://doi.org/10.1016/j.jff.2019.103636
**** Zhenghui, L.; Wenxing, H.; Yan, W.; Jihong, Z.; Xiaojun, X.; Lixin, G.; Mengshan, L. Ensemble learning based on bi-directional gated recurrent unit and convolutional neural network with word embedding module for bioactive peptide prediction. Food Chem. 2025, 468, 142464. https://doi.org/10.1016/j.foodchem.2024.142464
Reviewer 3 Report
Comments and Suggestions for Authors
The manuscript presents a timely and potentially impactful analysis of goat casein polymorphism and peptide bioactivity. With minor improvements to focus, methodology clarification, and interpretation, it will be suitable for publication.
- While the introduction offers a useful overview of milk protein polymorphism and bioactive peptides, the main novelty of the study is not yet clearly emphasized. It would benefit the reader to more explicitly define whether the innovation lies in the genotype-specific bioactivity prediction, in the use of specific casein alleles, or in the methodological integration of sequence and enzymatic analysis. A concise objective statement at the end of the Introduction would provide greater focus.
- The manuscript describes the in silico hydrolysis using trypsin and chymotrypsin A, but more information is needed on how cleavage sites were assigned. Were BIOPEP’s default rules used? Were overlapping peptides considered? A short clarification would improve transparency and help ensure reproducibility for other researchers employing this approach.
- As the peptide bioactivities are predicted based on sequence motifs matched to known fragments, there are inherent limitations to their functional interpretation. It would be advisable to briefly acknowledge the predictive nature of this approach and suggest that experimental confirmation is required to validate these findings.
- The manuscript ends rather abruptly. Adding a sentence or two to suggest logical next steps, such as synthesis and in vitro testing of the most promising peptides, or genotyping of breeds for functional trait selection would help underline the broader value of the work.
Author Response
"Please see the attachment."

Round 2
Reviewer 2 Report
Comments and Suggestions for Authors
The images are still blurred (I hope the editors can help in this issue).
Most of my concerns have been addressed, except this one:
Reviewer: Ln. 275, are there any weights for different components when computing Sigma A. Why?
Authors: No, as now is specified (please see L305-L309) ∑A, is an unweighted value considering all identified bioactive fragments.
I think the question "Why?" is not answered. I mean, will the application of some weights (because there are no specific reasons that all components weigh the same here) changes the conclusion of your work?
But frankly speaking, I am using the standard of theoretical chemistry to criticize your bioinfo work... This is not very fair. Since most of the questions of mine have been answered and I rejected your work previously mainly because I think its not a typical Molecules paper. I think I will let the editors to decide then if the work is in the scope of the journal.
Author Response
"Please see the attachment."
